# Determination of Antioxidant, Antimicrobial Activity, Heavy Metals and Elements Content of Seaweed Extracts

**DOI:** 10.3390/plants11111493

**Published:** 2022-06-01

**Authors:** Natália Čmiková, Lucia Galovičová, Michal Miškeje, Petra Borotová, Maciej Kluz, Miroslava Kačániová

**Affiliations:** 1Institute of Horticulture, Faculty of Horticulture, Slovak University of Agriculture, Tr. A. Hlinku 2, 949 76 Nitra, Slovakia; n.cmikova@gmail.com (N.Č.); l.galovicova95@gmail.com (L.G.); 2AgroBioTech Research Centre, Slovak University of Agriculture, Tr. A. Hlinku 2, 94976 Nitra, Slovakia; michal.miskeje@gmail.com (M.M.); petra.borotova@uniag.sk (P.B.); 3Department of Bioenergy, Food Technology and Microbiology, Institute of Food Technology and Nutrition, University of Rzeszow, 4 Zelwerowicza St, 35601 Rzeszow, Poland; kluczyk82@op.pl

**Keywords:** seaweed, algae, antimicrobial activity, antioxidant activity, heavy metals, PCA

## Abstract

The aim of the research was to determine the antioxidant and antimicrobial activity, determination of chemical elements and heavy metals in seaweed extracts of wakame, arame, dulse, laminaria, kombu, and hijiki. Antioxidant activity was determined by DPPH method and the activity ranged from 0.00 to 2641.34 TEAC. The highest antioxidant activity was observed in kombu (2641.34 TEAC) and arame (2457.5 TEAC). Antimicrobial activity was analyzed by disk diffusion method and MIC method. Three G^+^ bacteria (*Staphylococcus aureus*, *Enterococcus faecalis*, *Bacillus subtilis*), three G^-^ bacteria (*Salmonella enterica*, *Pseudomonas aeruginosa*, *Yersinia enterocolitica*), and four yeasts (*Candida tropicalis*, *C. krusei*, *C. glabrata*, *C. albicans*) were used as model organisms. The size of inhibition zones ranged from 0.00 to 8.67 mm. The minimum inhibitory concentrations of the selected seaweeds ranged from MIC_50_ 98.46 (MIC_90_ 100.25) to MIC_50_ 3.43 µL/mL (MIC_90_ 5.26 µL/mL). The content of selected elements was determined in seaweed samples by ICP-OES. The chemical composition of the algae showed differences between species and the presence of heavy metals. Arsenic, cadmium, and aluminum were confirmed. All seaweed samples contained arsenic ranging from 6.6546 to 76.48 mg/kg. Further investigation of seaweeds is needed to identify the active substances present in the algae that are responsible for antioxidant and antimicrobial activity. This study was carried out to evaluate the antimicrobial and antioxidant activity of extracts from five commonly consumed seaweeds for their ability to inhibit selected microorganisms and to determine the health risk due to heavy metals content. Our study contributes to the evidence that seaweeds have antimicrobial and antioxidant activity and seaweed extracts have for pharmacological applications.

## 1. Introduction

Seaweeds, or algae, are a large autotrophic diversified group [1]. Algae are predominantly aquatic photosynthetic organisms [2]. They are used in the food industry, cosmetic industry, agriculturally as fertilizers or as hydrocolloids used in agar production [2]. Seaweed is traditionally used as a food in Asian countries and is a valuable source of bioactive compounds [3]. Seaweed cultivation requires minimal external nutrient inputs and thus enables year-round biomass production. Seaweeds can be produced without arable land, freshwater, or fertilizers [4]. Seaweed biodiversity has recently attracted a lot of attention from researchers to find bioactive compounds that could contribute to the growth of the blue economy [5]. Marine organisms are considered as a potential source of biologically active compounds, but at the same time their chemical composition can provide biologically active compounds for drug development [6]. Algae produce a considerable number of compounds that are considered as secondary metabolites [7]. These metabolites are synthesized mainly at the end of the growth phase. Seaweeds contain a wide range of secondary metabolites that perform multiple functions. Moreover, due to their diverse biological activity, including their antibiotic properties, they show potential for human use [8]. Algae contain compounds, such as carotenoids, phenolic compounds, phycobiliprotein pigments, polysaccharides, and unsaturated fatty acids, which are considered as biologically active substances that affect antioxidant activity, anticancer activity, antimicrobial activity against bacteria, viruses, fungi, organic fertilizers, and also have bioremediation potential [9]. Seaweeds are one of the main producers of the marine ecosystem, found in almost all parts of coastal areas around the world [10].

Nowadays, the consumption and interest in seaweeds are increasing due to their natural composition. They contain about 80–90% water and the dry matter contains 50% carbohydrates, 1–3% lipids, and 7–38% minerals. The protein content varies and is highly variable (10–47%) with a high proportion of essential amino acids. Their low fat and high protein content make them interesting to the public and at the same time make them a food with very few calories [11]. Algae can be grown sustainably, and are a source of polar molecules, such as pigments and phenolic compounds, that have demonstrated antimicrobial potential [12].

Seaweeds are also important due to the presence of various minerals, such as Na, K, Ca, Mg, Fe, Zn, Mn, etc. Algae are also an important source of calcium. The calcium content varies from 7% in the dry matter to 25–34%. In addition, seaweeds also contain high concentrations of fluoride, 19.17–53.70 mg/g. Fluoride is considered an important chemical element for healthy teeth and bone integrity [12]. Although seaweeds contain significant amounts of biologically active compounds, some seaweeds show a high affinity for heavy metals [13]. Their detection is also used as a biomonitor for heavy metal pollution in rivers, seas, and oceans worldwide. Overall, algae and seaweed biomass can be used to sustainably remove heavy metals from wastewater [14]. The presence of heavy metals is dependent on environmental parameters and on the area (salinity, temperature, pH, light, nutrient concentrations, oxygen, etc.), and on structural differences between algal species [15]. Metal concentrations in seaweeds are generally low in summer due to rapid growth and thus high concentrations and dilution of accumulated metals, and conversely, they are especially high in winter when metabolic processes slow down [16].

The aim of the research was to analyze and determine the biological activity of seaweed extracts on selected Gram-positive, Gram-negative bacteria and yeasts. Antioxidant activity, antimicrobial activity by disk diffusion method, minimum inhibitory concentration, and analyses of the chemical composition of the algae were carried out. The content of heavy metals present in the seaweed extracts was also analyzed. Our study contributes to the evidence that seaweeds have antimicrobial and antioxidant activity and can inhibit the growth of microorganisms, as well as to the potential of seaweed extracts for pharmacological applications.

## 2. Results

### 2.1. Antioxidant Activity of Seaweeds

The antioxidant activity was determined by the DPPH method. The results were converted to percent inhibition and expressed according to the calibration curve as 1 μg Trolox per 1 g sample (TEAC) and 1 μg Trolox per 1 mg sample (TEACb) of algal extract. The free radical scavenging activity of the seaweed extracts ranged from 0.00 to 2641.34 TEAC of the sample (Table 1). The highest antioxidant activity was observed in kombu (2641.34 TEAC) and arame (2457.5 TEAC) seaweeds and on the contrary, the lowest activity was detected in dulse seaweed (56.31 TEAC) and even an absence of activity was determined in laminaria seaweed.

### 2.2. Antimicrobial Activity and Minimal Inhibition Concentration of Kombu

The seaweed kombu extract exhibited the highest antimicrobial activity on Gram-positive G^+^ bacteria, *Enterococcus faecalis*, producing an inhibition zone of 8 ± 1 mm, which can be considered as a strong antimicrobial activity (Figure 1). On another G^+^ bacteria, *Bacillus subtilis*, the kombu extract had a moderate effect (5.33 ± 0.58 mm). On all bacteria^-^, the extract had weak activity where inhibition zones were formed from 2.67 ± 0.58 to 4.00 ± 1.00 mm. On *Candida glabrata* and *Candida krusei*, the extract had moderate activity (7.33 ± 0.58 mm; 6.33 ± 0.58 mm) and on the remaining *Candida albicans* and *Candida tropicalis*, the extract from kombu had weak antimicrobial activity.

Kombu seaweed in our study showed inhibitory effects on all the studied microorganisms. They were least effective against yeasts with MIC 50 and MIC 90 values ranging from 24.46 to 12.53 µL/mL and 26.52 to 14.53 µL/mL (Figure 2). The extract was most effective in inhibiting the growth of microorganisms, especially against G^+^ bacteria, with MIC 50 and MIC 90 values ranging from 6.23 to 3.43 µL/mL and 8.36 to 5.26 µL/mL. These values confirm the results of antimicrobial efficacy against microorganisms obtained by the disk diffusion method.

### 2.3. Antimicrobial Activity and Minimal Inhibition Concentration of Laminaria

*Laminaria japonica* seaweed extract had weak antimicrobial activity on two G^+^ bacteria (*S. aureus*, *B. subtilis*) and on *E. faecalis*, the activity was strong activity (Figure 3). Laminaria seaweed extract showed moderate antimicrobial activity with zone of inhibition of 5.33 ± 0.58 mm. Laminaria extract had weak activity (*Y. enterocolitica*, *P. aeruginosa*) and on *S. enterica*, the extract had strong antimicrobial activity with zone of inhibition of 8.67 ± 0.58 mm. On yeasts (*C. glabrata*, *C. albicans*, *C. krusei*) the laminaria extract had moderate antimicrobial activity with zones of inhibition ranging from 5.67 to 7.33 mm. The inhibitory effect against *C. tropicalis* was weak with inhibition zones of 4.33 mm.

The values of MIC 50 and MIC 90 confirm the results obtained from the disk diffusion method (Figure 4). The MIC 50 ranged from 98.46 to 3.43 µL/mL and MIC 90 ranged from 100.25 to 5.26 µL/mL. The extract of the alga *L. japonica* was most effective against *C. krusei*, with minimum inhibitory concentrations of MIC 50 of 3.43 µL/mL (MIC 90 5.26 µL/mL). On the contrary, it was least effective against G^-^
*S. enterica* (MIC 50 98.46 µL/mL, MIC 90 100.25 µL/mL) and against G^+^ bacteria *E. faecalis* (MIC 50 98.46 µL/mL, MIC 90 100.25 µL/mL).

### 2.4. Antimicrobial Activity and Minimal Inhibition Concentration of Wakame

The wakame seaweed extract had weak antimicrobial activity on all G^+^ bacteria with inhibition zone sizes ranging from 3.33 to 4.67 mm (Figure 5). Similarly, the extract had weak inhibitory activity on two G^-^ bacteria (from 0.67 to 2.67 mm). The extract had moderate antimicrobial activity on *Yersinia enterocolitica*, with inhibition zones of 5.33 mm. The wakame extract was more effective on yeasts compared to bacteria. Algae had strong inhibitory effect on *Candida albicans* (inhibitory zones of 8.33 mm). It had moderate activity on the remaining three yeasts where inhibition zones ranging from 5.67 to 6.67 mm.

Algae wakame or *Undaria pinnatifida* had similar minimum inhibitory concentrations against G^+^, G^−^ bacteria, and yeasts. The MICs ranged from MIC 50 54.32 (MIC 90 55.18) to MIC 50 6.23 µL/mL (MIC 90 8.36 µL/mL) for the selected microorganisms (Figure 6). The wakame algae extract was most effective against *Bacillus subtilis* where MIC 50 6.23 µL/mL and MIC 90 8.36 µL/mL were the minimum inhibitory concentrations achieved. On the contrary, it was least effective against G^+^
*Enterococcus faecalis* (MIC 50 54.32 µL/mL, MIC 90 55.18 µL/mL).

### 2.5. Antimicrobial Activity and Minimal Inhibition Concentration of Dulse

The dulse seaweed (*Palmaria palmata*) belongs to the red seaweeds. We determined the antimicrobial activity using disk diffusion method. During the analysis, seaweed dulse did not achieve strong inhibitory activity for any microorganism (Figure 7). On yeasts, the extract had weak antimicrobial activity (from 3.67 ± 0.58 to 4.67 ± 0.58 mm) except for *Candida albicans*, where the size of inhibition zones reached 7.33 ± 0.58, which we consider as moderate activity. Moderate activity was achieved in G^+^ bacteria *Enterococcus faecalis* and in two G^-^ bacteria (*Yersinia enterocolitica*, *Pseudomonas aeruginosa*). For the other microorganisms, dulse seaweed extract achieved only weak antimicrobial activity.

*P. palmata* achieved demonstrable antimicrobial effects on all the microorganisms used in our analysis. The MIC values ranged from MIC 50 54.32 (MIC 90 55.18) to MIC 50 6.23 µL/mL (MIC 90 8.36 µL/mL) (Figure 8). The highest efficacy was achieved by the extract against the G^-^ bacteria *Pseudomonas aeruginosa* (MIC 50 6.23 µL/mL, MIC 90 8.36 µL/mL) and the lowest against the yeast *Candida krusei* (MIC 50 54.32 µL/mL, MIC 90 55.18 µL/mL).

### 2.6. Antimicrobial Activity and Minimal Inhibition Concentration of Hijiki

The antimicrobial activity of the seaweed *Sargassum fusiforme/Hizikia fusiformis* also called hijiki was determined using the disk diffusion method. This seaweed showed the lowest inhibitory activity on all microorganisms among all seaweed extracts. Moderate and strong activity was not present on G^+^, G^−^ bacteria, or microscopic filamentous fungi. Zero activity was observed for G^+^ bacteria, namely, *Staphylococcus aureus* (Figure 9). The radius of inhibition zones was determined from 0 to 3.67 mm.

Seaweed hijiki (*S. fusiforme*) showed inhibitory effects on all the microorganisms studied in our work. Among all the algae studied, hijiki had the weakest antimicrobial activity. It was the least effective against G^+^ bacteria with MIC 50 ranging from 98.46 to 54.32 µL/mL (MIC 90 100.25 to 55.18 µL/mL) (Figure 10). The extract showed the highest efficacy for growth inhibition against yeast with MIC 50 values ranging from 98.46 (MIC 90 100.25) to MIC 50 6.23 µL/mL (MIC 90 8.36 µL/mL). For G^-^ bacteria the MICs ranged from MIC 50 98.46 to 54.32 µL/mL (MIC 90 100.25 to 55.18 µL/mL). These values confirm the results that hijiki algae extract showed the weakest antimicrobial activity.

### 2.7. Antimicrobial Activity and Minimal Inhibition Concentration of Arame

The arame seaweed (*Eisenia bicyclis*) extract, like the hijiki seaweed extract, showed weak antimicrobial activity (Figure 11). For the G^+^ bacteria *Staphylococcus aureus*, the extract showed zero activity. Moderate and strong activity was not present against either microorganism. The diameters of the inhibition zones ranged from 0.00 to 3.33 mm.

An extract of the seaweed arame (*E. bicyclis*) showed antimicrobial activity. For Gram-positive G^+^ bacteria *Staphylococcus aureus*, *Enterococcus faecalis*, the extract showed the weakest antimicrobial activity with MIC 50 54.32 µL/mL (MIC 90 55.18 µL/mL) (Figure 12). For *Candida albicans* the extract showed the highest efficacy with MIC 50 6.23 µL/mL (MIC 90 8.36 µL/mL). Against other microorganisms, the extract of arame showed MIC 50 ranged from 25.42 to 12.46 µL/mL (MIC 90 from 27.36 to 14.53 µL/mL).

### 2.8. Determination of Chemical Elements and Heavy Metals in Seaweeds

Chemical elements, minerals and heavy metals present in selected dried seaweed samples were analyzed. Twenty-two essential elements were analyzed, and the values are expressed in units of mg/kg. The first element analyzed was silver, which was present in red dulse seaweed and brown arame seaweed at values ranging from 0.06 to 0.14 mg/kg. Aluminum was present in all seaweed samples ranging from 21.30 to 274.58 mg/kg with the highest value of aluminum measured in the alga laminaria. Arsenic was also present in all samples from 6.66 to 76.48 mg/kg. It is known that seaweeds contain abundant minerals, such as potassium, sodium, calcium, iron, magnesium, iodine, phosphorus, and zinc (Table 2). The highest amount of calcium was present in hijiki seaweed— 8853.82 mg/kg. Cadmium was also present in all the selected seaweeds except red dulse seaweed. Cobalt was present in only two algae, namely, hijiki and kombu. Chromium was also quantified in our work but its presence was not observed. In sea algae, copper (from 0.68 to 2.08 mg/kg), iron (from 11.44 to 78.44 mg/kg), potassium (from 12,412.15 to 29,609.10 mg/kg), lithium (from 0.20 to 1.53 mg/kg), magnesium (from 2472.84 to 9545.75 mg/kg), manganese (from 1.48 to 7.62 mg/kg), sodium (from 9430.59 to 15,349.30 mg/kg), zinc (from 5.45 to 9.13 mg/kg), and strontium (from 17.33 to 586.64 mg/kg) were determined. In our seaweed samples, we did not detect the presence of lead, antimony, and selenium, and molybdenum was present only in dulse seaweed.

### 2.9. Determination of Chemical Elements and Heavy Metals by Principal Component Analysis

Principal component analysis (PCA) partitioned the variability into six components but the most evident are PC1 and PC2. The PC1 (*x*-axis) accounts for 83% of the variability and the PC2 (*y*-axis) accounts for 11.7% of the variability. PC1 was clearly associated with total potassium and sodium (Figure 13), which were present in the highest content in the dulse and kombu algae samples and the lowest potassium content was present in the arame algae (Figure 14). Based on PC1, we can state that dulse, kombu, wakame, and also hijiki algae are similar (Figure 14). The factor that mainly influences PC1 is the potassium and sodium content, which allows us to claim that they have similar contents of these chemical elements. However, the laminaria and arame algae have considerably more variability in PC1 (Figure 13), and thus contain demonstrably less potassium and sodium. PC2, which presents 11.7% of the variation, is shown on the *y*-axis. For PC2, the major parameters are mainly the elements magnesium and calcium (Figure 13). This shows that hijiki and wakame have the highest content of these elements and dulse algae has the lowest content (Figure 14). The elements potassium, sodium, magnesium, and calcium show the most variation or differences in chemical elements and heavy metals content in the seaweeds we tested. Algae kombu and arame achieved the highest antioxidant activity (60.11% and 46.64% DPPH). On the basis of PCA statistics, it can be argued that they have a similarity in PC2, namely, similar magnesium and calcium content. However, on the basis of PC1 they had one of the highest variabilities and thus the highest potassium and magnesium content. Algae kombu and arame achieved the highest antioxidant activity (60.11% and 46.64% DPPH). On the basis of PCA statistics, it can be argued that they have a similarity in PC2, namely, similar magnesium and calcium content. However, on the basis of PC1 they had one of the highest variabilities and thus the highest potassium and magnesium content, where kombu algae contained the most of these elements and arame the least. Kombu contained the highest content of the elements responsible for algal variability, namely, K, Na, Mg, and Ca, among all algae and also achieved the strongest antioxidant activity. Hijiki and wakame have similar variability in both PC1 and PC2, similar content of the chemical elements studied, and their antioxidant activity was also very similar (9.09% and 11.38% of DPPH).

## 3. Discussion

Seaweeds are known for their high antioxidant activity, but this activity can be influenced by several factors, such as growth conditions, harvesting period, or the method of activity determination itself. Seaweed extracts were a dark green color, which also influences the determination of antioxidant activity. One factor influencing antioxidant activity is the form of algae processing. In a study by Amorim et al. [17], they compared the change in bioactive compounds and activity of kombu and wakame seaweeds in fresh and cooked state. Seaweed wakame in dry state showed significantly lower activity (5351.6 ± 2201.30 mg Tx/kg) compared to fresh wakame (12,263.18 ± 1657.67 mg Tx/kg). In their study they claimed that the antioxidant activity decreases after heat treatment of seaweeds. The effects of the pressure and pH changes, and by salting or freezing on antioxidant activity were addressed by del Olmo et al. [18]. The results indicate that the change in pressure, addition of salt and freezing have an effect on their biological activity. As the date of harvest and length of storage increased, the antioxidant activity decreased. Wang et al. [19] focused on the antioxidant activity of fractions extracted from *Laminaria japonica* by varying concentrations. They did not analyze the plant part but only fractions from it. Antioxidant activity ranged from 20 to 80%. The research confirmed that the extracts exhibit strong proton donating ability and can serve as free radical inhibitors or scavengers that can act as primary antioxidants, which is confirmed by our results. Yan et al. [20] compared the antioxidant activity of seaweeds. The seaweed *Hijiki fusiformis* achieved the highest activity, up to 65%. The second most effective algae for free radical scavenging were the wakame algae (*U. pinnatifida*) with an activity of 51.1%. The antioxidant activity of seaweeds is also influenced by the type of solvent used for the extract. Ismail et al. [21] prepared extracts of kombu, wakame, nori, and hijiki seaweeds and compared the antioxidant activity. The study shows that one of the most important factors affecting the antioxidant activity is the method of preparation of the extracts. Further analysis of the extracts with different solvents and comparison of their efficacy is needed.

Cai et al. [22] evaluated fractions from *L. japonica* as a natural preservative. The research showed that the petroleum ether, dichloromethane, ethyl acetate, and n-butyl alcohol fractions had obvious antimicrobial effects against *E. coli*, but the aqueous fraction had no antimicrobial effect. Kim et al. [23] focused on extracts from the alga laminaria and confirmed several biological activities. Polysaccharides of *L. japonica* have confirmed inhibitory effects on *E. coli* and *S. aureus*. A recent study reports that hot water extracts from laminaria also inhibit *A. actinomycetemcomitans*, *S. aureus*, and *E. coli*. However, the bioactive component in the ethanolic extracts of *L. japonica* that have antibacterial effects is still unclear. In our research, samples of *Laminaria japonica* from two producers, laminaria and kombu algae, were analyzed. These samples also differed from each other in antimicrobial activity (kombu from 1.33 ± 0.58 to 8.00 ± 1.00 and laminaria from 0.00 ± 0.00 to 8.67 ± 0.58). The results may have been mainly influenced by the origin of the seaweed and the growth conditions, where one was obtained from Japan and the other sample of laminaria was from China. The antimicrobial activity of the essential oil from the seaweed *L. japonica* was investigated in a study by Patra et al. [24]. They compared different fractions of essential oil from laminaria and evaluated which fraction has the strongest antimicrobial activity. The oil had a high fatty acid content with strong antioxidant and antimicrobial activity. The differences in the results and the strength of the antimicrobial activity may have been mainly influenced by the fact that they did not use the algae extract but the essential oil, which itself has inhibitory activity. In a study by Patra et al. [24] the effect of wakame algae essential oil on the inhibition of microorganisms was investigated. The essential oil was extracted from the brown edible seaweed *U. pinnatifida*. Essential oil, which also contains other components that may have been involved in the antimicrobial activity. Phull et al. [25] investigated the antibacterial activity of fucoidan derived from the seaweed *U. pinnatifida*. Their data suggested that maximum zone of inhibition of bacterial growth against *S. aureus* (15.67 ± 0.76 mm) and fungal growth against *A. fumigatus* (11.83 ± 1.01 mm) among other microorganisms. Our results showed lower antimicrobial activity, indicating lower concentration of fucoidan in the samples, which may indicate that fucoidan is the substance present in algae that is responsible for strong antimicrobial activity. Oumaskour et al. [26], in a study, investigated the antimicrobial activity of red seaweeds. The antimicrobial activity of the algae may have been influenced by the growth environment, time of collection, and different preparation of the extracts, which may have caused differences in our results compared to the present study. The antimicrobial activity of red seaweeds using different extraction solvents was tested against typical microorganisms that threaten the shelf life and safety of food products in a study by Abu-Ghannam and Rajauria [27]. The results showed comparable or even stronger antimicrobial efficacy especially when lipophilic extracts were used. These findings suggest that the activity of the purified compound against *L. monocytogenes* was almost 10-fold higher than that of the crude extract, supporting the role of purification in enhancing the overall biological activity. Li et al. [28] investigated the antioxidant and antimicrobial activity of polysaccharides from *S. fusiforme*. The inhibitory effects of the modified polysaccharides on the tested G^+^ bacteria (*S. aureus* and *B. subtilis*) were stronger (from 11.10 to 18.76 mm) than the effects on G^-^ bacteria (*E. coli*, *P. aeruginosa*, and *Salmonella* spp.). This study thus confirms our findings that hijiki seaweed has zero or very weak antimicrobial activity. *S. fusiforme* was analyzed in a study by El Shafay et al. [29]. The study confirms and demonstrates the importance of choosing a suitable method for algae extraction and also choosing a suitable solvent. The methanolic extract of *E. bicyclis* in a study by Eom et al. [30] showed antimicrobial activity against *Staphylococcus aureus* according to the disk diffusion test. In a study by Kim et al. [31], they evaluated the antifungal activity of phlorotannins extracted from the seaweed arame against *Candida albicans*. This study reaffirms that it is necessary to fractionate the seaweed extract and examine the components for antimicrobial activity individually, as there may not be sufficient concentration of the active ingredient in pure extracts.

Algae kombu and its minimum inhibitory concentration were investigated in a study by Kim et al. [23] on ethanol extracts against oral microbial species. Ethanol extract has demonstrable antimicrobial activity (62.5–500 μg/mL), which may have influenced the potency of the extract. The kombu algae extract also showed antimicrobial potency on our selected microorganisms as well as on the microorganisms selected in this study. In the work of Cai et al. [22], the ethanolic extract *of L. japonica* showed obvious antimicrobial effects against *S. aureus*, *B. subtilis*, *E. coli*, *P. vulgaris*, *E. aerogenes*, and *C. tropicalis*. Among the tested pathogens, the ethanolic extract of *L. japonica* showed the strongest antimicrobial activity against *E. coli* (260 μg/mL). However, the work of Cai et al. [22] also demonstrates the antimicrobial efficacy of kombu seaweed extract. Hoe et al. [32] investigated the antibacterial effects of seaweeds. The study confirms and demonstrates the importance of selecting a suitable method for algae extraction along with the choice of a suitable solvent. Liu et al. [33] evaluated the antibacterial activity of modified compounds derived from seaweed extracts against *E. coli* and *S. aureus*. The results obtained from the study indicate the possible potential of seaweeds and their modified compounds as new avenues for obtaining bioactive compounds. Patra et al. [34] analyzed the antioxidant and antibacterial properties of essential oil extracted from the edible seaweed *U. innatifida*. The results compared to ours could have been influenced by several factors, such as the use of a different form of algae, namely, in the form of essential oil, which also contains other constituents that may have been involved in the antimicrobial activity. Hellio et al. [35] investigated the inhibition of marine bacteria by macroalgae extracts They showed no minimum inhibitory concentration on G^+^ bacteria. The study demonstrates antimicrobial ability mainly against G^-^ bacteria, as demonstrated by our results. Garcia-Oliveira et al. [36] addressed macroalgae as an alternative source of nutrients and compounds with bioactive potential. *P. palmata* had the highest antibacterial potential against G^-^ bacteria *Escherichia coli*. The results confirm the antimicrobial activity of seaweed dulse. Li et al. [28] investigated the antioxidant and antimicrobial activity of polysaccharides from *S. fusiforme*. The individual polysaccharides have weaker antimicrobial activity than the whole algae. Prasedya et al. [37] tested *S. cristaefolium* extract for antimicrobial activity against *Staphylococcus aureus* (1.302 µg/mL). These results indicate compounds (fucoxanthin, known as a carotenoid) that may have an effect on antimicrobial activity, but need to be tested for other microorganisms. Kim et al. [38] performed a focused research of the in vitro antibacterial activity from the edible brown alga *E. bicyclis*. The results of this study suggested that the extract has anti-listerial activity. Kim et al. [31] isolated fucofuroeccol-A from the edible seaweed *E. bicyclis* to restore antifungal activity. The results obtained in this study suggest that phlorotannins, derived from *E. bicyclis*, may be a potential source for the development of an antifungal agent.

According to Shalaby [9], the brown seaweed *U. pinnatifida* and the red seaweed *Chondrus crispus* can be used as a dietary supplement to help meet the recommended daily intake of certain minerals, macronutrients, and trace elements. Mania et al. [39] determined the highest levels of inorganic arsenic in hijiki algal species while other algal samples gave an average inorganic concentration. All contaminants were found to be in non-compliance with the legal limits in a study by Almela et al. [40]. Choi et al. [41] determined the mineral content of dried seaweed (*L. japonica*). As a result of comparing the concentrations of four major metallic elements (Na, K, Ca, Mg) and three types of trace metallic elements (Fe, Zn, P), the Fe and Na contents were found to be high in Wando seaweed. Truus et al. [42] analyzed the brown alga *Fucus vesiculosus*. Attention was focused on the determination of Pb, Cd, Cu, Zn, Mn, Cr, and As in algal samples, by atomic absorption spectrometry. Arsenic and cadmium contents were determined directly from seawater. Bhat et al. [43] determined concentrations of minerals and heavy metals in 23 plants. Most of the plant materials were rich in some essential minerals, such as Na, K, Ca, Fe, Mg, Cu, Mn, and Zn, which are known to be beneficial to health. The data on the levels of heavy metal contaminants in plants highlight the necessity to control the quality and safety of seaweeds as well. In this study by Asensio et al. [44], 33 seaweeds were analyzed. The results indicate that the levels of Cd, Ni, and Pb in some samples were relevant and may pose a risk to consumers. Hg concentrations were generally the lowest detected in the samples. The seaweed hijiki (*Hizikia fusiforme*) contained a significant fraction of As. Chen et al. [45] addressed the distribution of metals and metalloids in dried seaweeds. Seaweeds of different geographic origin had a heterogeneous distribution of elements. This suggests that there is a low health risk for these elements for normal people due to seaweed intake. The main toxic metal present in seaweeds in the study by Paz et al. [46] was Al (38.9 mg/kg of dry seaweed). Regarding the origin, the highest concentrations of Al, Cd, and Pb were found in Asian seaweeds, which may be due to the high level of industry.

## 4. Materials and Methods

### 4.1. Tested Seaweed

Dried seaweed purchased from a commercial chain was used to prepare the extracts. The seaweeds wakame/*U. pinnatifida* (nutrition facts per 100 g: energy: 817 kJ/199 kcal, fat: 0.3 g, carbohydrates: 0 g, protein: 1.8 g, salt: 0.547 g, storage dry), arame/*E. bicyclis* (nutrition facts per 100 g: energy: 181 kJ/81.9 kcal, fat: 0.3 g, carbohydrates: 0 g, protein: 0.8 g, fiber: 68.9 g, salt: 0.377 g, dry storage), kombu/*L. japonica* (nutrition facts per 100 g: energy: 7.1 kJ/1 kcal, fat: 0.10 g, carbohydrates: 0.20 g, protein: 0.1 g, fiber: 34.9 g, salt: 0.02 g, dry storage) originate from Japan. Hijiki/*S. fusiforme* (nutrition facts per 100 g: energy: 1288 kJ/304 kcal, fat: 2.1 g, carbohydrates: 65.0 g, protein: 6.6 g, fiber: 47.0 g, salt: 9.3 g, dry storage) also originated from Japan, laminaria/*L. japonica* (nutrition facts per 100 g: energy: 22.6 kJ/5.4 kcal, fat: 0.2 g, carbohydrates: 0.0 g, protein: 0.9 g, fiber: 29.0 g, salt: 0.02 g, dry storage) originates in China. The seaweed dulse/*P. palmata* (nutrition facts per 100 g: energy: 1078 kJ/255 kcal, fat: 0.3 g, carbohydrates: 38.0 g, protein: 18.1 g, fiber: 14.2 g, salt: 0.05 g, dry storage), which was obtained from the Atlantic Ocean off Spain, was analyzed. All algae samples were dried and vacuum packed in 2021 and they were stored at room temperature.

### 4.2. Microorganisms

G^+^ bacteria (*Bacillus subtilis* CCM 1999, *Staphylococcus aureus* subsp. *aureus* CCM 2461, *Enterococcus faecalis* CCM 4224), G^-^ bacteria (*Pseudomonas aeruginosa* CCM 3955, *Yersinia enterocolitica* CCM 7204, *Salmonella enterica* subsp. *enterica* ser. *enteritidis* CCM 4420) and 4 yeasts (*Candida krusei* CCM 8271, *Candida albicans* CCM 8261, *Candida tropicalis* CCM 8264, *Candida glabrata* CCM 8270), were obtained from the Czech collection of microorganisms (Brno, Czech Republic).

### 4.3. Preparation of Extracts

Dried sea algae samples were mechanically crushed into smaller pieces and diluted with ethanol at the ratio of 5 g of crushed algae and 45 mL of 96% ethanol (Sigma Aldrich, Schnelldorf, Germany). The samples were macerated for 7 days in the dark on a shaker. After one week of maceration, extracts were prepared using a vacuum rotary evaporator (Witegvapor, Witeg Labortechnik, Wertheim, Germany). Evaporation was carried out in a water bath at 40 °C with a rotation speed of 20 rpm. The resulting extract was diluted with 0.1% DMSO (1 mL) and the yield of each seaweed was calculated. Extracts of seaweeds were stored at 4 °C in the dark.

### 4.4. Determination of Antioxidant Activity

The antioxidant activity of seaweeds was analyzed using the DPPH (2,2-diphenyl-picrylhydrazyl) (Sigma Aldrich, Schnelldorf, Germany) radical assay on a 96-well microplate. As a first step, a DPPH solution of 0.025 g/L dissolved in 99% methanol (Uvasol for spectroscopy, Merck, Darmstadt, Germany) was adjusted to an absorbance of 0.8 at 515 nm, which was measured on a Glomax (Promega Inc., Madison, WI, USA). Furthermore, 5 µL of sample and 195 µL of DPPH solution were pipetted onto the microplate. The reaction solutions thus prepared were incubated under continuous shaking at 1000 rpm for 30 min in the dark. The determination of antioxidant activity was calculated as the percentage inhibition of DPPH according to the formula:(1)$$\%\;\mathrm{inhibition}=\;\frac{\left(A0-\mathrm{AA}\right)}{A0}\;\times \;100$$
where A0 = absorbance of DPPH and AA = absorbance of the sample.

Furthermore, the antioxidant activity was calculated relative to a standard reference substance Trolox (Sigma Aldrich, Schnelldorf, Germany) dissolved in methanol to a concentration in the range of 0–100 µg/mL. Finally, the total antioxidant activity was expressed according to the calibration curve as 1 µg Trolox per 1 g algal extract sample (TEAC).

### 4.5. Disk Diffusion Method

The principle of the disk diffusion method is to determine the sensitivity of the microorganisms and the size of the inhibition zones to our chosen extracts. Blank disks (Oxoid, Basingstoke, UK) made of 6 mm filter paper were used and placed on a Petri dish with agar where the inoculum was added. Bacteria, both G^−^ and G^+^, were incubated in Mueller Hinton broth (MHB, Oxoid, Basingstoke, UK) at 37 °C for 24 h. For yeasts, Sabouraud’s dextrose broth (SDB, Oxoid, Basingstoke, UK) was used and incubated at 25 °C for 24 h. The microorganisms prepared were adjusted to 0.5 McFarland standard using a densitometer which corresponds to 1.5 × 10^8^ colonies per milliliter of forming units (CFU). A Petri dish with Mueller Hinton agar (MHA) was inoculated with 100 μL of bacterial culture, Sabouraud’s dextrose agar (SDA) was used for yeast and clean disks were placed on the surface and 10 μL of extract was pipetted on top. Samples were incubated for 24 h at 37 °C for bacteria and 25 °C for yeast. After this time, the inhibition zones (radius) were measured, and the antimicrobial activity was statistically evaluated by the disk diffusion method in triplicate. Two antibiotics (cefoxitin 30 µg/disk, gentamicin 30 µg/disk; Oxoid, Basingstoke, UK) and one antibiotic (fluconazole 30 µg/disk; Oxoid, Basingstoke, UK) were used as positive controls for G^-^, G^+^ bacteria, and yeasts. Disks with distillated water were used as a negative control. Samples were analyzed in triplicate and the mean (*n* = 3) was applied to the results and the standard deviation was calculated. Based on the zone of inhibition, the algal extracts were categorized by weak (0–5 mm, *), moderate (5–8 mm, **), and strong (>8 mm, ***) antimicrobial activity.

### 4.6. Minimum Inhibitory Concentration (MIC)

The minimum inhibitory concentration, also called MIC method, is a method that determines the minimum concentration that can inhibit the growth and reproduction of microorganisms. This method was carried out in microplates where suitable medium for the microorganisms was present. MHB liquid culture medium was used to determine the minimum inhibitory concentration of seaweed extracts for bacteria and SDB was used for yeasts. The microorganisms were cultured in the given culture media for 24 h at an optical density of 1.5 × 10^8^ CFU/mL. Furthermore, 15 μL of the culture was pipetted into a 96-well plate where 30 μL of liquid culture medium was present, followed by the addition of 30 μL of seaweed extract. Concentrations of algal extract were added ranging from 0.2 to 400 μL/mL per well. In addition to the minimum inhibitory concentration, we also determined the negative control and the maximum growth control. Pure MHB (SDB for yeasts) with extract was used as negative control and MHB (SDB for yeasts) with microbial culture without extract was used as positive growth control. A Glomax plate spectrophotometer (Promega Inc., Madison, WI, USA) was used for MIC determination at an absorbance of 570 nm. MIC values were determined based on the concentration of extract at which the absorbance was less than or equal to the negative control. The MIC value results (concentration that caused 50% and 90% inhibition of bacterial growth) were determined by probit analysis.

### 4.7. Determination of Heavy Metals

The contents of selected elements (Ag, Al, As, Ba, Ca, Cd, Co, Cr, Cu, Fe, K, Li, Mg, Mn, Mo, Na, Ni, Pb, Sb, Se, Sr, and Zn) were determined in selected seaweed samples by inductively coupled plasma optical emission spectrometry (ICP-OES). All chemicals used in the methodology were highly pure. The biological material, crushed seaweed, was stored in a dark place and at room temperature. The weight of the experimental samples ranged from 0.5 to 2.0 g. Samples were mineralized in an Ethos UP high-power microwave system (MilestoneSrl, Sorisole, BG, Italy) in a solution of 5 mL HNO3 ≥ 69.0% (Trace SELECT, Honeywell Fluka, Morris Plains, NJ, USA), 1 mL H_2_O_2_ ≥ 30%, for trace analysis (Sigma-Aldrich, St. Louis, MI, USA), and 2 mL ultrapure water (18.2 MΩ/cm; 25 °C, Synergy UV, Merck Millipore, France). The method consists of heating and cooling phases. During the heating phase, the samples were heated to 200 °C for 15 min and this temperature was maintained for a further 15 min. During the cooling phase, the samples underwent 15 min of active cooling to reach a temperature of 50 °C. Digests were filtered through VWR 454 quantitative filter paper (particle retention 125 μm) (VWR International, Leuven, France) into volumetric flasks and filled with ultrapure water to a volume of 50 mL. Elemental analysis (Ag, Al, As, Ba, Ca, Cd, Co, Cr, Cu, Fe, K, Li, Mg, Mn, Mo, Na, Ni, Pb, Sb, Se, Sr, and Zn) was performed using an inductively coupled plasma-optical emission spectrometer (ICP OES 720, Agilent Technologies, Melbourne, Australia (M PtyLtd.)) with an axial plasma configuration and an SPS-3 autosampler (Agilent Technologies, Basel, Switzerland). Multi-element standard solution V for ICP (Sigma-Aldrich Production GmbH, Busch, Switzerland) was used in the experiment. The detection limits (μg/kg) of the trace elements measured were as follows. The validity of the whole method was verified using a certified reference material (CRM-ERM CE278 K, Sigma Aldrich Production GmbH, Busch, Switzerland). The obtained results were statistically processed in GraphPad Prism 6.01 (GraphPad Software Incorporated, San Diego, CA, USA) [47].

### 4.8. Statistical Data Processing

Inhibition zones in the disk diffusion method were measured in triplicate (*n* = 3). Their mean and standard deviation were processed using Microsoft Excel software. SAS software version 8 was used to process the MIC data. The results of MIC values (concentration that caused 50% and 90% inhibition of microorganism growth) were determined by probit analysis. The total antioxidant activity was expressed according to the calibration curve as 1 μg Trolox per 1 mL of extract sample (TEAC). Furthermore, the results were expressed as percentage inhibition and the activity was converted to the standard reference substance Trolox. The contents of chemical elements and heavy metals were statistically processed using GraphPad Prism 6.01 software (GraphPad Software Incorporated, San Diego, CA, USA).

## 5. Conclusions

The present work is focused on the determination of antioxidant and antimicrobial activity, minimum inhibitory concentrations, chemical elementals composition, and heavy metals content of extracts of wakame, arame, dulse, laminaria, kombu, and hijiki seaweeds. The results of this work suggest that the algae contain substances that are capable of inhibiting the growth of microorganisms and have antioxidant activity. The most effective seaweed extract for inhibition of G^+^
*E. faecalis* was from kombu seaweed, against G^-^
*S. enterica* was from laminaria seaweed extract, and against *C. albicans* was from wakame seaweed extract. The minimum inhibitory concentration was also determined and the most effective extract against *S. aureus* was from kombu, laminaria extract against *C. krusei*, dulse extract against *P. aeruginosa*, hijiki extract against *C. albicans*, and *C. krusei*, arame extract against *C. albicans*, and wakame seaweed extract against *B. subtilis*. The chemical composition of the seaweeds was diverse, and the species differed from each other. The presence of trace elements and important minerals, such as magnesium, calcium, zinc, selenium, and iron, were confirmed. Seaweeds absorb heavy metals from the environment and can therefore be used as an indicator of environmental pollution. All seaweed samples contained arsenic. Therefore, we anticipate that advances in algae biotechnology will have a major impact on the current industrial landscape and will spur the emergence of a sustainable and efficient algae-based bioeconomy that will be key to overcoming the challenges and constraints that conventional agriculture will face in the years to come.

## Figures and Tables

**Figure 1 plants-11-01493-f001:**
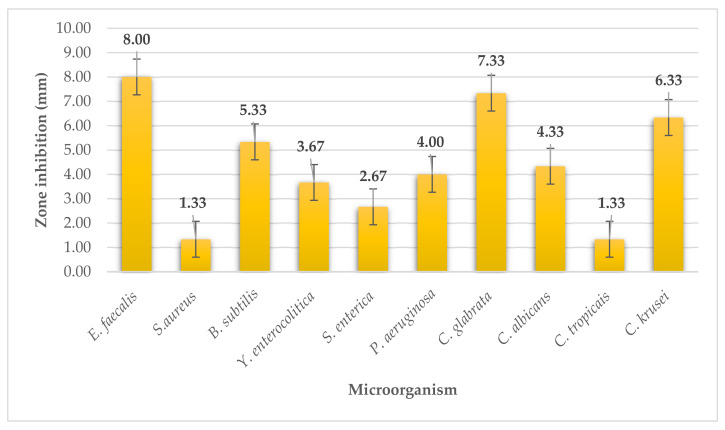
Antimicrobial activity of kombu.

**Figure 2 plants-11-01493-f002:**
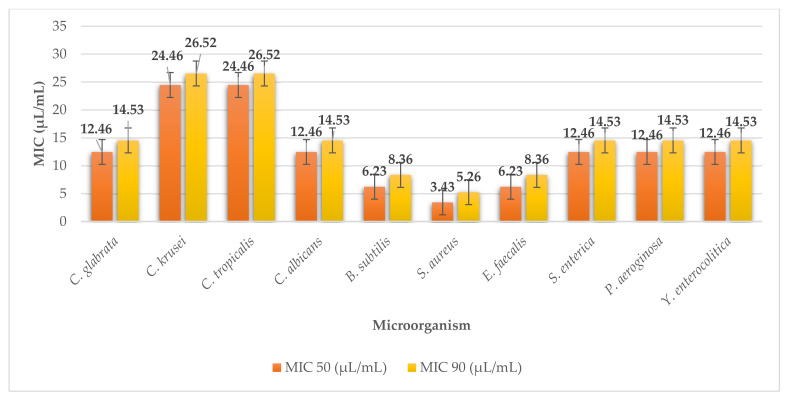
MIC 50 and MIC 90 of kombu.

**Figure 3 plants-11-01493-f003:**
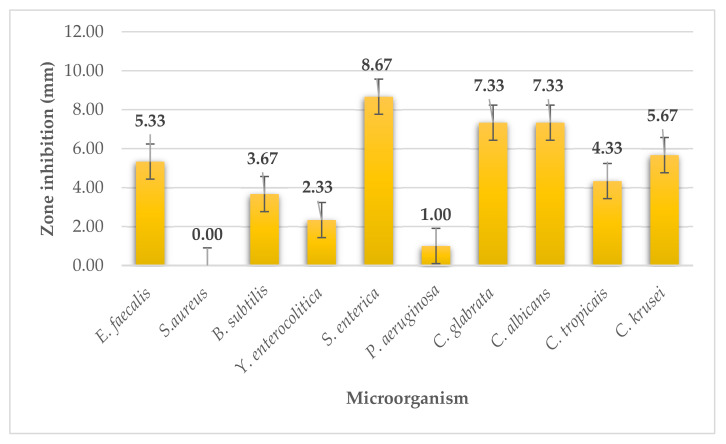
Antimicrobial activity of laminaria.

**Figure 4 plants-11-01493-f004:**
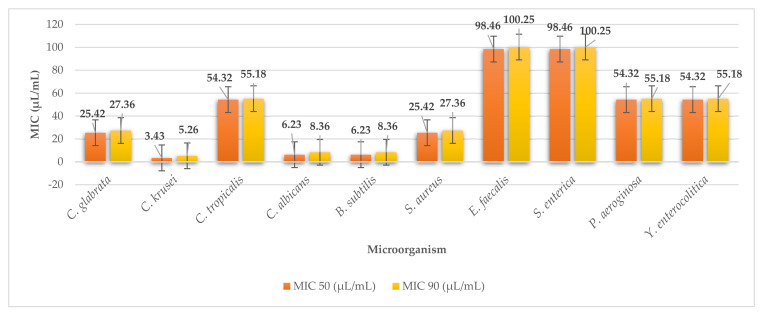
MIC 50 and MIC 90 of laminaria.

**Figure 5 plants-11-01493-f005:**
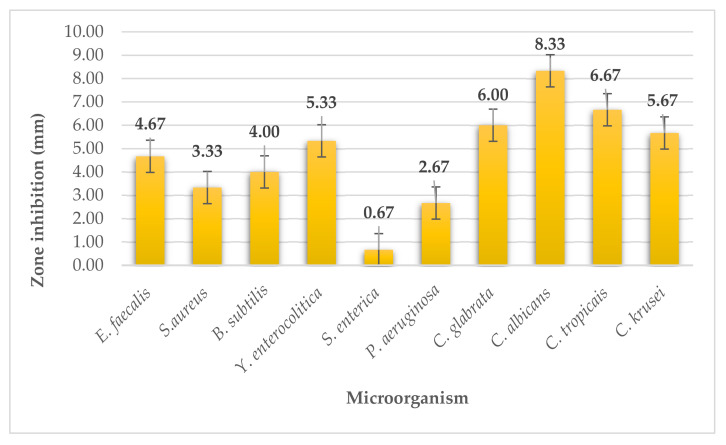
Antimicrobial activity of wakame.

**Figure 6 plants-11-01493-f006:**
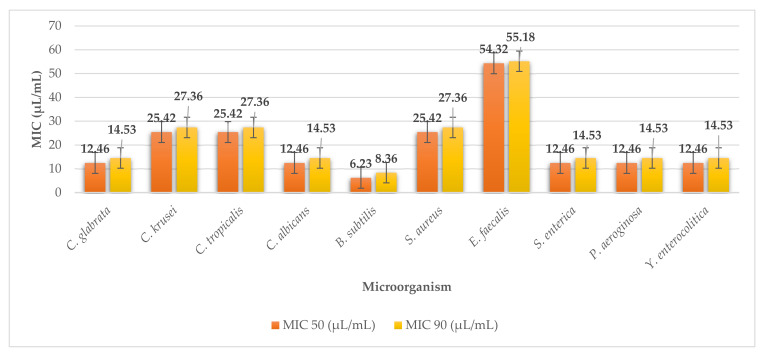
MIC 50 and MIC 90 of wakame.

**Figure 7 plants-11-01493-f007:**
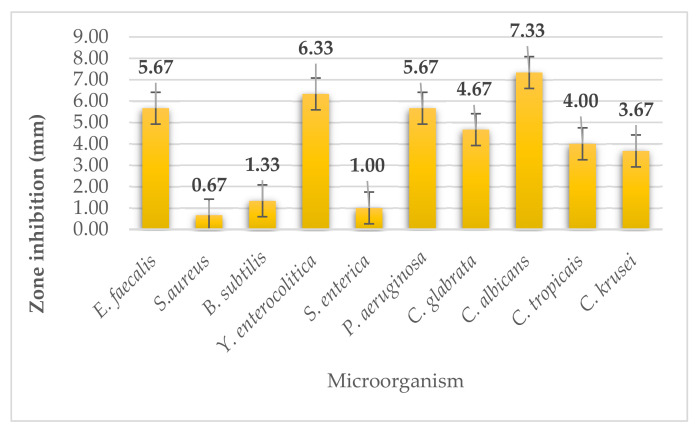
Antimicrobial activity of dulse.

**Figure 8 plants-11-01493-f008:**
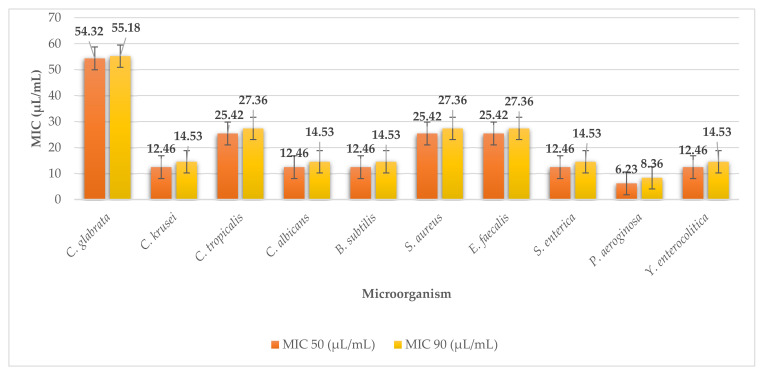
MIC 50 and MIC 90 of dulse.

**Figure 9 plants-11-01493-f009:**
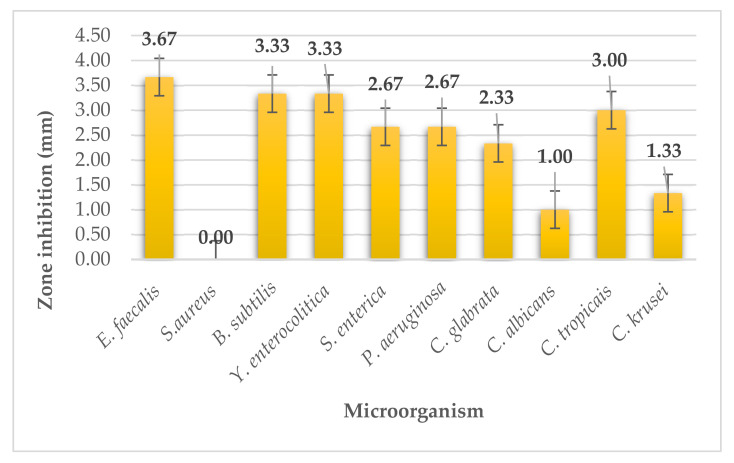
Antimicrobial activity of hijiki.

**Figure 10 plants-11-01493-f010:**
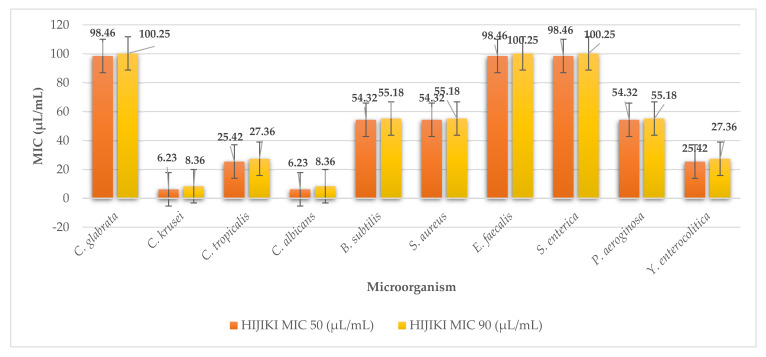
MIC 50 and MIC 90 of hijiki.

**Figure 11 plants-11-01493-f011:**
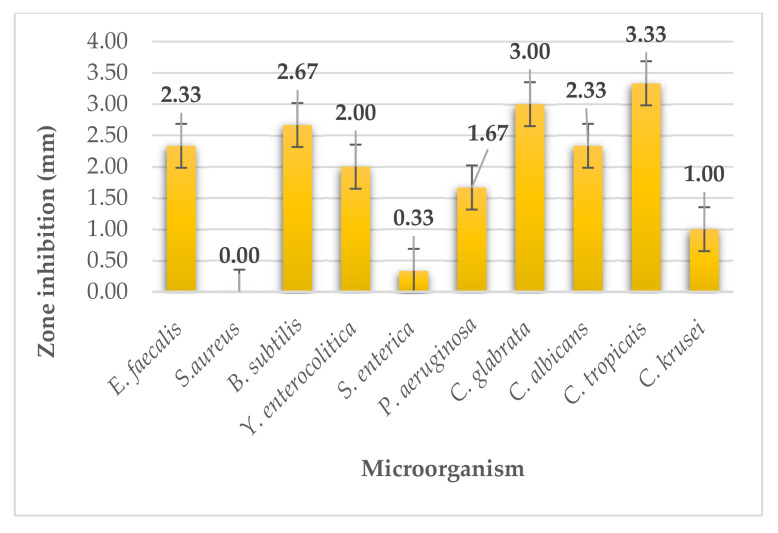
Antimicrobial activity of arame.

**Figure 12 plants-11-01493-f012:**
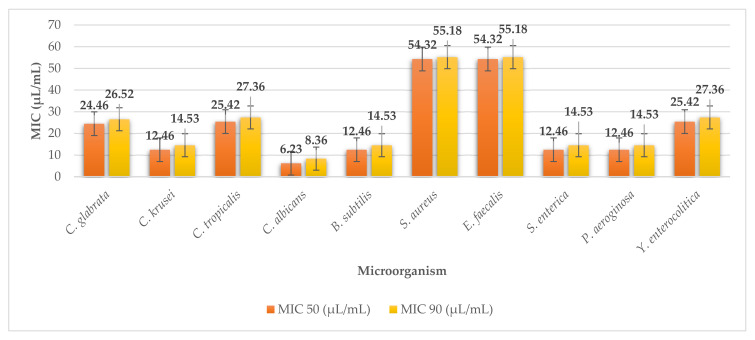
MIC 50 and MIC 90 of arame.

**Figure 13 plants-11-01493-f013:**
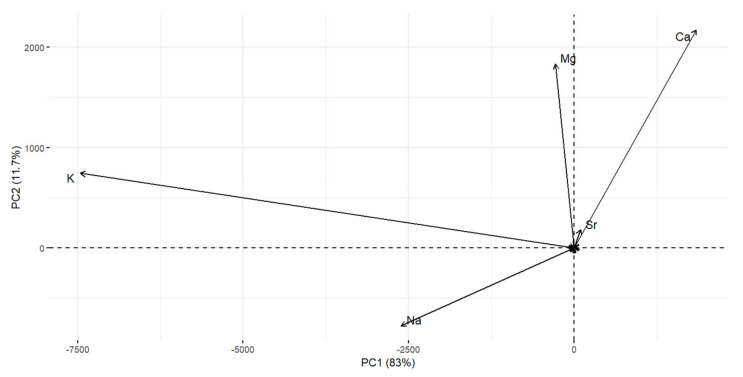
Principal component analysis of elements.

**Figure 14 plants-11-01493-f014:**
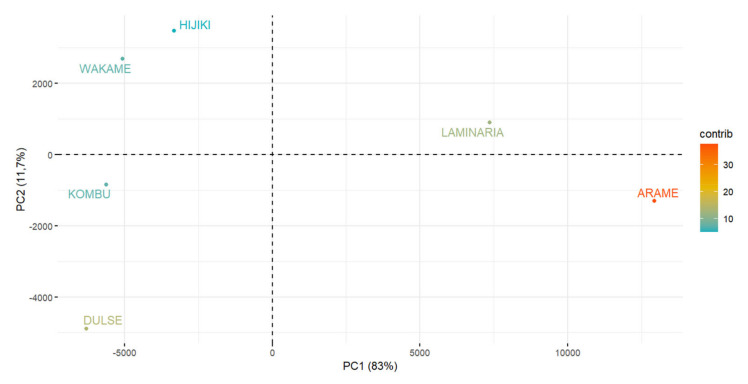
Principal component analysis of seaweeds.

**Table 1 plants-11-01493-t001:** Antioxidant activity of seaweeds.

Samples	DPPH (%)	TEAC	TEACb
kombu (*Laminaria japonica*)	60.11	2641.34	2.64
laminaria (*Laminaria japonica*)	-	-	-
hijiki (*Sargassum fusiforme*	9.09	755.26	0.76
wakame (*Undaria pinnatifida*)	11.38	170.01	0.17
dulse (*Palmaria palmata*)	13.08	56.31	0.06
arame (*Eisenia bicyclis*)	46.64	2457.5	2.46

**Table 2 plants-11-01493-t002:** Chemical elements and heavy metals of seaweeds.

Elements	Absorbance (nm)	Hijiki *(S. fusiforme)* (mg/kg)	Wakame *(U. pinnatifida)* (mg/kg)	Laminaria *(L. japonica)* (mg/kg)	Dulse (*P. palmata*) (mg/kg)	Arame (*E. bicyclis*) (mg/kg)	Kombu *(L. japonica)* (mg/kg)
Ag	328.07	ND	ND	ND	0.06	0.138	ND
Al	167.02	21.30	46.37	274.58	72.95	21.02	56.59
As	188.98	76.48	23.14	27.95	6.66	18.54	66.76
Ba	455.40	4.36	6.90	13.08	0.14	8.10	3.14
Ca	315.89	8853.82	6002.98	8184.28	1244.91	8374.60	4670.21
Cd	226.50	0.69	0.14	0.53	ND	0.43	0.36
Co	228.62	0.22	ND	ND	ND	ND	0.32
Cr	267.72	ND	ND	ND	ND	ND	ND
Cu	324.75	1.16	0.68	1.71	1.24	2.08	0.78
Fe	234.35	11.44	43.33	78.44	35.51	15.87	18.58
K	766.49	29,581.10	29,181.05	18,390.50	29,609.10	12,412.15	29,609.10
Li	670.78	0.48	1.53	0.32	0.44	0.20	0.56
Mg	383.83	5553.73	9545.75	5838.97	2472.84	4385.13	5116.46
Mn	257.61	5.16	4.45	14.78	7.62	1.89	1.48
Mo	204.60	ND	ND	ND	0.40	ND	ND
Na	589.59	11,222.05	15,349.30	9430.59	15,346.05	9455.70	15,349.30
Ni	231.60	2.28	ND	0.73	0.84	ND	ND
Pb	220.35	ND	ND	ND	ND	ND	ND
Sb	206.83	ND	ND	ND	ND	ND	ND
Se	196.03	ND	ND	ND	ND	ND	ND
Sr	407.77	586.64	482.04	454.33	17.33	505.57	252.14
Zn	206.20	6.40	7.63	9.13	7.24	8.44	5.45

## Data Availability

Data is contained within the article.

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
