# Peer review of "Determination of Antioxidant, Antimicrobial Activity, Heavy Metals and Elements Content of Seaweed Extracts"

_plants, 2022, doi:10.3390/plants11111493_

Round 1

Reviewer 1 Report

Dear Authors,

Your the manuscript entitled "Chemical Composition and Biological Activity of Seaweed" by Natália Čmiková, Lucia Galovičová, Michal Miškeje, Petra Borotová, Maciej Kluz and Miroslava Kačániová may be of interest to readers of the journal.

In recent years, many researchers have turned their attention to the great diversity of marine organisms. Algae are considered one of the richest sources of biologically active ingredients with powerful biological activity. Algae or marine macroalgae are an important source of bioactive ingredients that find many and varied applications in industry such as food and dairy products, pharmaceuticals, medical, cosmetic, nutraceutical, etc.

The authors should pay special attention to the "Introduction" section because the sources you cite are very old. I don't think you've taken your goal seriously enough. I have many new sources even from the current 2022.

In the section "Results" the authors have presented well the results obtained by them are clear and accurate. They are well presented in the relevant tables.

Regarding the "Discussion" part, the authors discussed the results accurately and clearly obtained from them for antioxidant activity.

In the section "Methods and materials" the authors have described well the methods used in this manuscript. In my opinion, any researcher could repeat the analyzes made by the authors for different species of algae.

The conclusion correctly summarizes the results obtained by the authors and their discussion.

And last but not least, as far as the "References" part is concerned, I would like to point out that the sources used by the authors are, in my opinion, very old. Of the 39 sources, only 14 are from the last 3-5 years. Let the authors pay special attention to the literature because there are many new sources on the topic they have worked on, even from the current year 2022.

I have no conflict of interest with the authors of this manuscript.

Author Response

Reviewer 1

Your the manuscript entitled "Chemical Composition and Biological Activity of Seaweed" by Natália Čmiková, Lucia Galovičová, Michal Miškeje, Petra Borotová, Maciej Kluz and Miroslava Kačániová may be of interest to readers of the journal.

In recent years, many researchers have turned their attention to the great diversity of marine organisms. Algae are considered one of the richest sources of biologically active ingredients with powerful biological activity. Algae or marine macroalgae are an important source of bioactive ingredients that find many and varied applications in industry such as food and dairy products, pharmaceuticals, medical, cosmetic, nutraceutical, etc.

Thank you, reviewer, for your insights, comments and questions about our article.

Point 1: The authors should pay special attention to the "Introduction" section because the sources you cite are very old. I don't think you've taken your goal seriously enough. I have many new sources even from the current 2022.

Response: It was added.

Point 2: In the section "Results" the authors have presented well the results obtained by them are clear and accurate. They are well presented in the relevant tables.

Response: The Authors are very grateful to the Reviewer.

Point 3: Regarding the "Discussion" part, the authors discussed the results accurately and clearly obtained from them for antioxidant activity.

Response: The Authors are very grateful to the Reviewer.

Point 4: In the section "Methods and materials" the authors have described well the methods used in this manuscript. In my opinion, any researcher could repeat the analyzes made by the authors for different species of algae.

Response: The Authors are very grateful to the Reviewer.

Point 5: The conclusion correctly summarizes the results obtained by the authors and their discussion.

Response: The Authors are very grateful to the Reviewer.

Point 6: And last but not least, as far as the "References" part is concerned, I would like to point out that the sources used by the authors are, in my opinion, very old. Of the 39 sources, only 14 are from the last 3-5 years. Let the authors pay special attention to the literature because there are many new sources on the topic they have worked on, even from the current year 2022.

Response: In introduction and references were new sources added.

Comments and suggestions have been incorporated into the article. Tables was changed to figures by the instruction of other reviewers.

Reviewer 2 Report

I have some suggestions for the article. These are in the body of the article attached to this platform.

Author Response

Reviewer 2

I have some suggestions for the article. These are in the body of the article attached to this platform.

Thank you, reviewer, for your insights, comments and questions about our article. All suggestion of reviewer was accepted.

Reviewer 3 Report

Article titled: Chemical Composition and Biological Activity of Seaweed.

This work is of interest. However, in my opinion it is suitable for publication after a major revision. Please, find my suggestions:

1) The title is too generic.

2) Abstract must be re-write. Please, insert more details on seaweed. Please, remove the sentence "Extracts were evaporated 13 using a vacuum evaporator and dissolved in 0.1% DMSO." I don't think it's useful in an abstract. Moreover, the originality of the study must be noted.

3) Introduction: The purpose and originality of the study must be noted.

4) The antioxidant activity was evaluated only with the DPPH test. Authors should test this activity with other assays.

5) Paragraph 4.1. must be improved. The identification must be inserted.

Author Response

Reviewer 3

Article titled: Chemical Composition and Biological Activity of Seaweed.

Thank you, reviewer, for your insights, comments and questions about our article.

This work is of interest. However, in my opinion it is suitable for publication after a major revision. Please, find my suggestions:

Point 1: The title is too generic.

Response: New title: Determination of antioxidant antimicrobial activity and heavy metal and elements content of seaweed extracts.

Point 2:  Abstract must be re-write. Please, insert more details on seaweed. Please, remove the sentence "Extracts were evaporated 13 using a vacuum evaporator and dissolved in 0.1% DMSO." I don't think it's useful in an abstract. Moreover, the originality of the study must be noted.

Response: Abstract was re-write.

Point 3: Introduction: The purpose and originality of the study must be noted.

Response: In the introduction, we stated the purpose and originality of the study.

Point 4:  The antioxidant activity was evaluated only with the DPPH test. Authors should test this activity with other assays.

Response: We did DPPH method that it is a basic method whose results are sufficient, at the same time the high arsenic content also affects the antioxidant activity, which affected the results of other methods we tried on our seaweed extracts. We did DPPH method that it is a basic method whose results are sufficient, at the same time the high arsenic content also affects the antioxidant activity, which affected the results of other methods we tried on our seaweed extracts.

Point 5:  Paragraph 4.1. must be improved. The identification must be inserted.

Response: 4.1 identification was added.

Reviewer 4 Report

Overall, a very good paper is submitted. The English is acceptable. The figures are well presented. References are adequate. It is good to have algae Latin names in Tables 1 and 8. 

Author Response

Reviewer 4

Overall, a very good paper is submitted. The English is acceptable. The figures are well presented. References are adequate. It is good to have algae Latin names in Tables 1 and 8. 

Thank you, reviewer, for your insights, comments and questions about our article. Algae Latin names are now in Tables 1 and 8.

Round 2

Reviewer 1 Report

Dear Authors,

I agree with the manuscript you have corrected, the added figures give a clearer idea of the results you have obtained.

Author Response

Reviewer 1

Dear Authors,

I agree with the manuscript you have corrected, the added figures give a clearer idea of the results you have obtained.

Thank you, reviewer, for your positive response.

Reviewer 3 Report

This work is now suitable for publication after a minor revision.

The style of Figures 3, 5, 7 , and 9 must be revised.

The titles of Figures 6, 8, and 10 should be smaller.

Author Response

Reviewer 3

This work is now suitable for publication after a minor revision.

Thank you, reviewer, for your insights, comments and questions about our article.

Point 1: The style of Figures 3, 5, 7 , and 9 must be revised.

Response: The style and formal points of figures were changed.

Point 2: The titles of Figures 6, 8, and 10 should be smaller.

Response: The title of figures were removed and all figures in manuscript were uniformed.